# Pericarditis and Takotsubo Syndrome—Diagnosis of Cardiac Complications of Post-Acute COVID-19 Syndrome in a 77-Year-Old Woman

**DOI:** 10.3390/diagnostics12102304

**Published:** 2022-09-24

**Authors:** Malgorzata Zalewska-Adamiec, Hanna Bachorzewska-Gajewska, Slawomir Dobrzycki

**Affiliations:** 1Department of Invasive Cardiology, Medical University of Bialystok, 15-089 Białystok, Poland; 2Department of Clinical Medicine, Medical University of Bialystok, 15-089 Białystok, Poland

**Keywords:** Takotsubo syndrome, pericarditis, COVID-19, SARS-CoV-2, post-acute COVID-19 syndrome, PACS

## Abstract

The SARS-CoV-2 virus infection most often takes the form of acute COVID-19 respiratory disease, but in some patients, it turns into acute COVID-19 syndrome after a few weeks. Cardiac complications occur in the form of acute and post-acute diseases and the most common are myocarditis, pericarditis, arrhythmias, and acute coronary syndromes or Takotsubo syndrome. Cardiovascular complications are often the cause of hospitalization and death in COVID-19 patients. We present the case of a 77-year-old woman who was admitted to the clinic with suspected myocardial infarction. Coronary arteriography revealed atherosclerotic wall lesions, and echocardiography showed a good contractility of the left ventricle and fluid in the pericardial sac. Pericarditis was diagnosed. In the following days, acute kidney damage was observed, and one hemodialysis session was performed. On the sixth day of hospitalization, a sudden cardiac arrest occurred, and the patient was resuscitated. The echocardiogaphy showed abnormal contractility of the left ventricular with the ejection fraction of 15%—Takotsubo image. After a few hours, a cardiac arrest occurred again, and the patient died.

**Figure 1 diagnostics-12-02304-f001:**
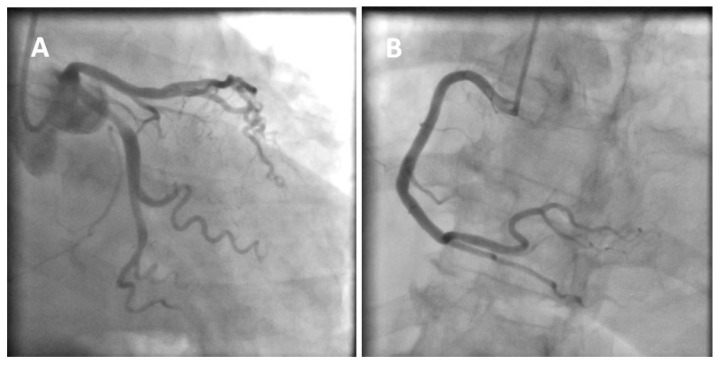
A 77-year-old obese woman with approximately 20 h of retrosternal pain was admitted to the clinic with suspected inferior STEMI infarction. The patient suffered from COVID-19 for four weeks before admission. On admission to the clinic, the patient reported chest pain. The ECG showed atrial fibrillation with a frequency of QRS about 100/min and ST-segment elevation in I, II, III, aVF, and V2-V6. The patient underwent coronary angiography, which revealed parochial atherosclerotic lesions in the coronary arteries (**A**,**B**) and, in the aortography, there were no signs of aortic dissection. The echocardiography showed normal valves function, good left ventricular systolic function with 58% ejection fraction (LVEF), and the presence of fluid in the pericardial sac, up to 10–11 mm behind the lower ((**C**), white arrow) and lateral ((**D**), yellow arrow) walls. The laboratory tests showed increased parameters of inflammation and normal levels of troponin and cardiac enzymes. Pericarditis was diagnosed and a treatment was started with colchicine, ibuprofen, enoxaparin, ceftriaxone, and levofloxacin. On the second day of hospitalization, the clinical condition of the patient deteriorated. There was an increase in renal parameters (creatinine—1.45 mg/dL, urea—100.5 mg/dL, and eGFR—37 mL/min) and atrial fibrillation with a fast QRS rate of 150/min. Pulmonary edema occurred, which was controlled by the intravenous infusion of furosemide and nitroglycerin. An echocardiogram was performed, which showed no significant changes. On the next day, the patient had a further deterioration in renal function with anuria. Hemodialysis was performed, followed by a return to diuresis. In the following days of hospitalization, the clinical condition of the patient improved. An echocardiographic examination was repeated in the patient every 1–2 days, confirming the good systolic function of the left ventricle and a constant amount of fluid in the pericardial sac. On the sixth day of hospitalization, the patient experienced a sudden cardiac arrest in the ventricular fibrillation (VF) mechanism. Resuscitation activities were undertaken, obtaining an atrial rhythm of about 90–100/min. In addition, the electrocardiogram showed a right bundle branch block and left bundle branch posterior bundle block (RBBB + LPH), QTc—510 ms. The patient was connected to a respirator and administered dobutamine, noradrenaline by continuous intravenous infusion, and amiodarone. The echocardiography revealed extensive abnormalities of contractility with akinesis of the apex, apical, and middle segments of all left ventricular walls, which may be consistent with Takotsubo syndrome (**E**,**F**). The ejection fraction was estimated at approximately 15%. There was a small amount of fluid in the pericardial sac. After a few hours, the patient returned to sudden cardiac arrest by VF and pulseless electrical activity (PEA). The resuscitation measures undertaken turned out to be ineffective, and the patient was pronounced dead. Patients with cardiovascular diseases are at risk of a severe course of COVID-19, as they are at risk of exacerbation of existing cardiological diseases and a higher risk of complications. Cardiological complications also often occur in COVID-19 patients with no previous history of cardiac diseases [1]. Pericarditis occurs in 3% of patients with COVID-19. The main symptom of pericarditis is retrosternal pain; ECGs show diffuse ST-segment elevations and echocardiography shows a small amount of fluid in the pericardial sac and elevated parameters of inflammation. The prognosis of patients is usually favorable [2]. The literature data on Takotsubo syndrome in COVID-19 is limited, but to date it has been noticed that a small percentage affects women, who constitute 50–60% of the groups of patients (90% before COVID-19). The more frequent stressor causing TS is the physical factor [3,4]. The diagnosis of Takotsubo syndrome in COVID-19 is significantly difficult and is often based on echocardiography and the probability of Takotsubo according to the InterTAK scale [5]. The prognosis of patients is unfavorable, and mortality is about 30–40%. Takotsubo syndrome with severe arrhythmias and acute heart failure was the direct cause of death of our patient. The patient’s condition required the use of inotropic drugs, which probably did not improve her prognosis. Currently, in TS with cardiogenic shock, the use of levosimendan is recommended and mechanical circulatory support should be considered (intra-aortic balloon pump, Impella or ECMO) [6,7]. Various cardiac complications in both acute COVID-19 and later post-COVID syndromes worsen the prognosis of these patients. The number of stress factors associated with falling ill with COVID-19 increases the risk of Takotsubo syndrome. The differences in Takotsubo syndrome in COVID-19, demonstrated to date, require confirmation in subsequent prospective studies and discussion on the verification of the criteria for diagnosing TS in patients with COVID-19.

## Data Availability

Original data supporting the reported results are available upon request from the authors.

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
