# Peer review of "Pericarditis and Takotsubo Syndrome—Diagnosis of Cardiac Complications of Post-Acute COVID-19 Syndrome in a 77-Year-Old Woman"

_diagnostics, 2022, doi:10.3390/diagnostics12102304_

Round 1

Reviewer 1 Report (Previous Reviewer 3)

It's a quite important case. More data should be added concerning the use of Aortic balloon implantation in order to improve the hemodynamic stability of the patient. The inotropic infusion is not appropriate practice

Author Response

Dear Reviewer

Thank you very much for your valuable attention. We agree that the use of traditional inotropic drugs in Takotsubo patients is not a good choice. Levosimendan would be better, but the patient's condition did not allow us to change. On the other hand, we did not use IABP due to the quick death of the patient (2 hours after the first SCA). The excerpt about inotropic drugs and mechanical circulatory support has been added to the manuscript and marked in green.

Best regards

Reviewer 2 Report (Previous Reviewer 4)

I'm satisfied of your clarifications.

Author Response

Dear Reviewer

Thanks a lot for your review.

Best regards

This manuscript is a resubmission of an earlier submission. The following is a list of the peer review reports and author responses from that submission.

Round 1

Reviewer 1 Report

1- The treatment indicated to the patient: Ceftriaxone and cyprofloxacine it is not justified in the context of Covid infection.

2- tha patients has too many comorbilities to asume that the pericarditis and the contractility abnormalities was dur to Sarscov 2 infection.

Author Response

Dear Reviewer

Thank you very much for your kind remarks.

1- The treatment indicated to the patient: Ceftriaxone and cyprofloxacine it is not justified in the context of Covid infection.

The patient received ceftriaxone and levofloxacin. It was indeed not the best set of antibiotics, but cefriaxone was applied to the county hospital and levofloxacin was added to the clinic. Currently, it was not an acute COVID and a decrease in inflammatory parameters was observed, therefore the antibiotics were not changed.

2- tha patients has too many comorbilities to asume that the pericarditis and the contractility abnormalities was dur to Sarscov 2 infection.

Before the SARS-Co2 infection, the patient was diagnosed only with hypertension and diabetes, therefore the most likely cause of pericarditis was a history of COVID-19 a month earlier. On the other hand, further complications (arrhythmias, pulmonary edema, renal failure and Takotsubo syndrome) occurred in the next days of hospitalization after the diagnosis of pericarditis.

All changes in the text were marked on red.

Yours sincerely

Malgorzata Zalewska-Adamiec

Reviewer 2 Report

Thank you for the interesting article. 

I would like to ask the authors to clarify two points: 

The diagnosis of Takotsubo in COVID-19 is significantly difficult-what difficulties do the authors want to highlight that would be specifically related to COVID

The differences in Takotsubo syndrome in COVID-19, demonstrated so far, require confirmation in subsequent prospective studies and discussion on the verification of the criteria for diagnosing TS in patients with COVID-19.-Takotsubo cardiomyopathy is often reversible, do the authors think that the prospective studies will always detect the patients with cardiomyopathy? 

Is there any prior ECHO to know the ejection fraction? as the low ejection fraction may not be associated with COVID only. It is not possible to comment on the effect of COVID unless we know about the prior cardiac function. 

Author Response

Dear Reviewer

Thank you very much for your kind remarks.

1. The diagnosis of Takotsubo in COVID-19 is significantly difficult-what difficulties do the authors want to highlight that would be specifically related to COVID.

We have a reasonable suspicion that, as in the case of cancer patients, as well as in the case of COVID patients, many cases of Takotsubo syndrome have not been diagnosed. Symptoms of acute COVID and Takotsubo symptoms are often similar (dyspnoea, chest pain), which may delay the diagnosis of TS. Patients with COVID are hospitalized in infectious diseases wards under a sanitary regime, where access to cardiological consultations and cardiological imaging tests, especially echocardiography, may be limited.

2. The differences in Takotsubo syndrome in COVID-19, demonstrated so far, require confirmation in subsequent prospective studies and discussion on the verification of the criteria for diagnosing TS in patients with COVID-19.-Takotsubo cardiomyopathy is often reversible, do the authors think that the prospective studies will always detect the patients with cardiomyopathy? 

We believe that prospective studies will not detect all cases of Takotsubo syndrome, but may improve the diagnosis of TS in COVID and thus also the prognosis of these patients.

3. Is there any prior ECHO to know the ejection fraction? as the low ejection fraction may not be associated with COVID only. It is not possible to comment on the effect of COVID unless we know about the prior cardiac function. 

On admission to the clinic, echocardiography showed good left ventricular systolic function with an EF of 58%.

All changes in the text were marked on red.

Yours sincerely

Malgorzata Zalewska-Adamiec

Reviewer 3 Report

It's an interesting case focused on a post COVID period.

The pericarditis was confirmed. We need more data concerning the second to seventh day. Which were the echo data in that period while the patient appeared pneumonic edema?

Which was the dose of the inotropic drugs ? In case of high doses of inotropic infusion please your comments.

Author Response

Dear Reviewer

Thank you very much for your kind remarks.

1. The pericarditis was confirmed. We need more data concerning the second to seventh day. Which were the echo data in that period while the patient appeared pneumonic edema?

Echocardiographic examination was repeated in the patient every 1-2 days, confirming good systolic function of the left ventricle, a constant amount in the pericardial sac (added in the manuscript). The pulmonary edema was most likely caused by an attack of atrial fibrillation, and echocardiography, except for the pericardial fluid, was normal.

2. Which was the dose of the inotropic drugs ? In case of high doses of inotropic infusion please your comments.

Low doses of inotropic drugs were used; dobutamine 10 μg/kg/min and noradrenaline 1 mg/h.

All changes in the text were marked on red.

Yours sincerely

Malgorzata Zalewska-Adamiec

Reviewer 4 Report

The paper describes an interesting case of COVID 19 cardiac complications that leads to the death of the patient. Every contribution to the understanding of this disease is useful to improve our knowledge of Covid 19, particularly if the case leads to a fatal outcome. Anyway the description lacks of some important data that could change interpretation of this clinical case. So could you please describe:

1)      Dosage of troponin and cardiac enzymes at the presentation in ER

2)      Which could be the cause of pulmonary edema occurred on 2nd day? Was an echocardiogram performed in the acute phase of pulmonary edema?

3)      Did you perform  cardiac MRI or, if this wasn’t possibile, an autopsy?

4)      If you don’t have these data how can you exclude the presence of a myocarditis associated to pericarditis as cause of the death instead of a second different illness as Takotsubo syndrome?

It would be interesting to know something more about thoughts and interpretations that surely the Heart team did after the death of this patient.

Round 2

Reviewer 3 Report

The authors completed the reviewer's suggestions.